# Anxiety and depression during pregnancy: Differential impact in cases complicated by preeclampsia and preterm premature rupture of membranes

**Yolanda Giménez[1]\*, Elena González[1], Francesc Fatjó[2], Aida Mallorquí[3], Sandra Hernández[1], Angela Arranz[1], Francesc Figueras[1]**

1 Department of Maternal-Fetal Medicine, BCNatal, Barcelona Center for Maternal-Fetal and Neonatal Medicine, Hospital Clinic, University of Barcelona, Barcelona, Catalonia, Spain, 2 University of Barcelona, Barcelona, Catalonia, Spain, 3 Clinical Health Psychology Section, Hospital Clinic, Barcelona, Catalonia, Spain

\* ygimenez@clinic.cat

## Abstract

### Background

Maternal mental health is crucial for the well-being of both the mother and the fetus. Obstetric complications have been linked to anxiety and depression during pregnancy. Among them, preeclampsia (PE) and preterm premature rupture of membranes (PPROM), are the more common causes of maternal admission. The aim of this study is to explore whether there is an increasing prevalence in the gradient of anxiety and depression among women with uncomplicated pregnancies, those admitted for PPROM, and those admitted for PE.

### Methods

A cross-sectional t study was conducted involving three groups of pregnant women consecutively attended: 1) women admitted with severe PE; 2) women admitted with PPROM; and 3) uncomplicated pregnancies. Participants completed validated questionnaires to measure anxiety (State-Trait Anxiety Inventory, STAI), depression (Edinburgh Postnatal Depression Scale, EPDS). Differences in median scores across the study groups were analysed by quantile regression, adjusted for gestational age at evaluation and the STAI-Trait score.

### Results

The analysis included 214 women: 106 with uncomplicated pregnancies, 55 with PPROM, and 53 with severe PE. A higher proportion of nulliparity and chronic hypertension was observed in women with preeclampsia. Significant trends across the study groups were observed for both depression and anxiety scores. Women with severe preeclampsia had higher scores on the State-Trait Anxiety Inventory-State (STAI-S) compared to those with

**Data availability statement:** All relevant data are within the manuscript and its Supporting Information files.

**Funding:** The author(s) received no specific funding for this work.

**Competing interests:** The authors have declared that no competing interests exist.

PPROM (27 vs. 24; p=0.049). The PPROM and PE groups showed significantly higher proportions of abnormal scores in STAI-S>30, EPDS>10, and EPDS>13 compared to uncomplicated pregnancies.

## Conclusions

Higher levels of anxiety and depression are present in women admitted in hospital for PPROM and severe PE. Compared to PPROM, severe PE is associated with higher scores of anxiety. The importance of screening and offering specific interventions for patients with PE is highlighted.

## Introduction

Pregnancy, childbirth and the postpartum period are challenging times in a woman's life. Several physiological, psychological and social changes make it a period of continuous adaptation susceptible to the development of mental health problems [1].

Depression and anxiety during pregnancy, with a prevalence of 10% to 20%, significantly contribute to maternal and perinatal morbidity and mortality [2,3]. These disorders not only affect the quality of life of pregnant women but can also lead to suboptimal parenting and postpartum depression [4,5]. Additionally, it is worth noting that suicide constitutes 20% of maternal mortality in the postpartum period [6]. Regarding the offspring, these disorders have been associated with the development of cognitive, behavioural, interpersonal, and emotional problems throughout their lives, as well as attention deficit hyperactivity disorder in childhood [7]. This represents a significant challenge for public health due to its adverse impact on maternal and child health, along with potential economic implications if not adequately addressed [8,9].

Symptoms of anxiety and depression tend to intensify in the third trimester of pregnancy [10,11]. Despite this, attention to emotional disorders during pregnancy remains limited, and most research has focused on the identification and treatment of postnatal emotional problems in mothers, most commonly postpartum depression [12].

Furthermore, there is evidence that levels of anxiety and depression increase as gestational risk rises [13], with a prevalence reaching 36% in high-risk women [14].

Obstetric complications affect around 10% of pregnancies, increasing the risk of anxiety and depression, especially in situations of preterm hospitalisation and fears associated with maternal health and premature birth [15–17]. Among the reasons for hospitalization, pre-eclampsia (PE) and preterm premature rupture of membranes (PPROM) stand out as the most frequent. They carry a significant impact on maternal and neonatal morbidity and mortality, accounting for 15% and 30% of the preterm deliveries, respectively. Preeclampsia (PE), affects approximately 3–5% of pregnancies, is distinguished from preterm premature rupture of membranes (PPROM) due to its unique impact on maternal health [18,19].

The treatment of PE involves maternal hospitalisation in critical and/or semi-critical care units, with timely delivery and the use of magnesium sulphate. Additionally, its pathogenesis, linked to placental hypoperfusion and ischemia, triggers a systemic inflammatory response that is considered, in part, a pathophysiological mechanism of postpartum depression [20]. These factors explain why PE has a special association with depression and anxiety [21–24].

The objective of this study is to explore whether there is an increasing prevalence gradient of anxiety and depression among women with uncomplicated pregnancies, those admitted for PPROM, and those admitted for PE.

## Materials and methods

### Study design

A cross-sectional t study was conducted involving three groups of pregnant women consecutively attended to at a referral hospital (Hospital Clinic de Barcelona) between December, 2016 and December, 2018. Participation in the study was voluntary, and informed consent was obtained from each participant. The study received approval from the Clinical Research Ethics Committee of Hospital Clinic de Barcelona (Reg. HCB/2016/0489).

### Study subjects

3 groups were defined:

A.  Consecutive pregnant women diagnosed with severe PE according to "International Society for the Study of Hypertension in Pregnancy" (ISSHP) criteria [25] who required admission to a high-dependency obstetric care unit.

B.  Consecutive pregnant women with PPROM diagnosed by the presence of overt hydrorrhea on vaginal examination or with positive rapid test (PAMG-1) in vaginal specimen that required admission to a high-dependency obstetric care unit or to the high-risk hospitalisation ward.

C.  Pregnant women without medical or obstetric complications, who attended routine antenatal care at the same institution during the study period, were recruited simultaneously alongside patients with complications.

All participants included in the study were women aged 18 years or older, who were pregnant with a gestational age ranging from $26^{+0}$ to $33^{+6}$ weeks, and with the presence of a viable fetus. Exclusion criteria were established as personal psychiatric history, including previous history of depression and/or anxiety; experience of physical, psychological or sexual abuse; language barrier; and occurrence of perinatal death.

### Study variables

The main outcomes assessed were the depression level score, measured by the "Edinburgh Postnatal Depression Scale" (EPDS) [26] and the anxiety level score, assessed by the anxiety questionnaire "State-Trait Anxiety Inventory" (STAI) [27].

Both scales were self-administered concurrently within the initial 72 hours of admission for the PE and PPROM groups. In the control group, self-administration occurred during the routine prenatal visit, conducted between weeks $26^{+0}$ and $33^{+6}$.

The EPDS scale comprises 10 items formulated in positive and negative terms. Participants indicated the intensity or frequency of their feelings and experiences in the last two weeks using a Likert-type scale ranging from "0" (No, not at all) to "3" (Yes, most of the time). The total scale score ranges from 0 to 30. A cut-off of ≥13, based on the validation study of the instrument in the general Spanish population [28], was used. This cut-off point is supported by a Cronbach's Alpha coefficient of 0.79, a sensitivity of 85%, and a specificity of 77%. Scores were interpreted as follows: no risk of depression for EPDS < 10; risk of depression between 10–12 points, and EPDS ≥ 13 indicating probable depression.

The STAI scale, consisting of 40 items, breaks anxiety down into two separate concepts: anxiety as a trait (STAI-T), which represents a relatively stable anxious propensity, and anxiety as a state (STAI-S), which reflects the transient emotional condition. The temporal scale of reference differs for the two dimensions: "right now, at this moment" for state anxiety (20

items) and "generally, most of the time" for trait anxiety (20 items). Each subscale consists of 20 items formulated in the affirmative and negative, using a 4-point Likert response system according to intensity (0= almost never/not at all; 1= somewhat/sometimes; 2= quite often; 3= very much/almost always). The total score on each subscale ranged from 0 to 60. A cut-off of ≥30, according to established norms, was applied to identify affirmative states of anxiety. The version used in this study corresponds to the Spanish adaptation of the "State-Trait Anxiety Inventory" by Spielberg, with a population mean for the state scale of 23.30 and internal consistency assessed by the Cronbach's Alpha coefficient, yielding a value of 0.93 [29].Participants first self-completed the state anxiety items and subsequently the trait anxiety items.

As control variables, the following were recorded: age at recruitment (years); origin (European/ non-European); educational level (no education/ primary/ secondary/ university; unemployment; household status (couple/ single); primiparous (no previous delivery beyond 20 weeks); previous miscarriage (pregnancy loss before 20 weeks); previous perinatal death (beyond 20 weeks); smoker in pregnancy (1 or more cigarettes/day at booking); chronic hypertension (blood pressure >140/90 measured on two occasions more than 12h apart before 20 weeks of pregnancy); dyslipidaemia (pre-pregnancy cholesterol > 200 mg/dL); maternal heart disease (congenital structural anomalies, acquired ischemic, valvular or arrhythmic conditions); diabetes (pre-gestational; gestational) according to "Spanish Diabetes and Pregnancy Group" (GEDE) criteria [30]; gestational age at inclusion (days); gestational age at delivery (days); birthweight (gr); fetal growth restriction (an estimated fetal weight below the 10th percentile according local standards); caesarean delivery; maternal hospitalisation (days); maternal hospitalisation in obstetric high-dependency unit (HDU); maternal hospitalisation days in HDU (days).

## Statistical analysis

We expected to recruit 50 women with preterm premature rupture of membranes (PPROM) and 50 with preeclampsia (PE) over the study period. An additional 100 uncomplicated pregnancies were intended to be included as a control group, resulting in a total planned sample size of 200 participants. To determine the adequacy of our sample size for detecting a trend in proportions across the study groups, we performed a power analysis using the linear-by-linear trend test. We assumed a significance level of 0.05 and three ordered exposure groups with expected proportions of 5%, 20%, and 30%. Using a trend test framework with equally spaced exposure levels, the estimated power for detecting a significant linear trend was 85.8%.

The normality of the distribution of continuous variables was evaluated by visual inspection of the stem-and-leaf plots and formally tested using the Shapiro-Wilk test. A descriptive analysis of the continuous variables was carried out with the mean, standard deviation (SD) and range. Categorical variables were characterised by absolute frequencies and percentages corresponding to each category.

Differences in the quantitative outcome variables were assessed using paired quantile regression of the medians (PPROM vs. uncomplicated pregnancies, PE vs. uncomplicated pregnancies, and PE vs. PPROM), adjusted for gestational age at evaluation and the STAI-Trait score. Trends across groups were univariately assessed using the Jonckheere–Terpstra test. Multivariately, the trend across the study groups was analysed using quantile regression of the medians, adjusting for gestational age at evaluation and the STAI-Trait score, where the study group was treated as a linear polynomial contrast.

Differences in the categorical outcome variables (STAI-S ≥ 30, EPDS ≥ 10 and EPDS ≥ 13) were assessed using paired logistic regressions (PPROM vs. uncomplicated pregnancies, PE vs. uncomplicated pregnancies, and PE vs. PPROM), adjusted for gestational age at evaluation

and gestational age at diagnosis; Trends across groups using the linear-by-linear test. Multi-variately, the trend across the study groups was analysed using logistic regression, adjusting for gestational age at evaluation, and the STAI-Trait score, where the study group was treated as a linear polynomial contrast.

The level of statistical significance adopted was p<0.05. All of these analyses were performed using SPSS 23.0 statistical software (IBM Corp., Armonk, NY).

## Results

A total of 233 women were approached. Of them, 19 (8.2%) were excluded for the following reasons: 15 (6.4%) voluntarily withdrew from the study; 3 were excluded because of the occurrence of a fetal death before the outcome measurements; and one additional woman was excluded because she was diagnosed after inclusion for a psychiatric condition. Thus, a total of 214 were included in the analysis: 106 uncomplicated pregnancies, 55 with PPROM and 53 with severe PE Fig 1.

Table 1 shows the baseline and perinatal outcomes by study group. Of note, regarding the baseline characteristics, the proportion of nulliparity and chronic hypertension was higher in women with preeclampsia. Concerning the perinatal outcomes, as expected, women with preeclampsia had a smaller baby and a higher incidence of fetal growth restriction.

Table 2 and Figs 2 and 3 display the anxiety and depression scores by study group. The PPROM and PE groups had significantly higher median STAI-S and EPDS scores than the uncomplicated pregnancies. Women with preeclampsia had higher STAI-S scores than women with PPROM (27 vs. 24; adjusted-p=0.049). There was a significant trend of higher STAI-S and EPDS scores across the study groups, which persisted significant after adjustment for gestational age at evaluation and STAI-T scores.

Table 3 details the proportion of abnormal anxiety and depression scores by study group. Of note, both the PPROM and PE groups had a statistically significant higher proportion of STAI-S≥30 and EPDS≥10 scores compared to the uncomplicated pregnancies. There was a significant trend of a higher proportion of abnormal STAI-S and EPDS scores across the study

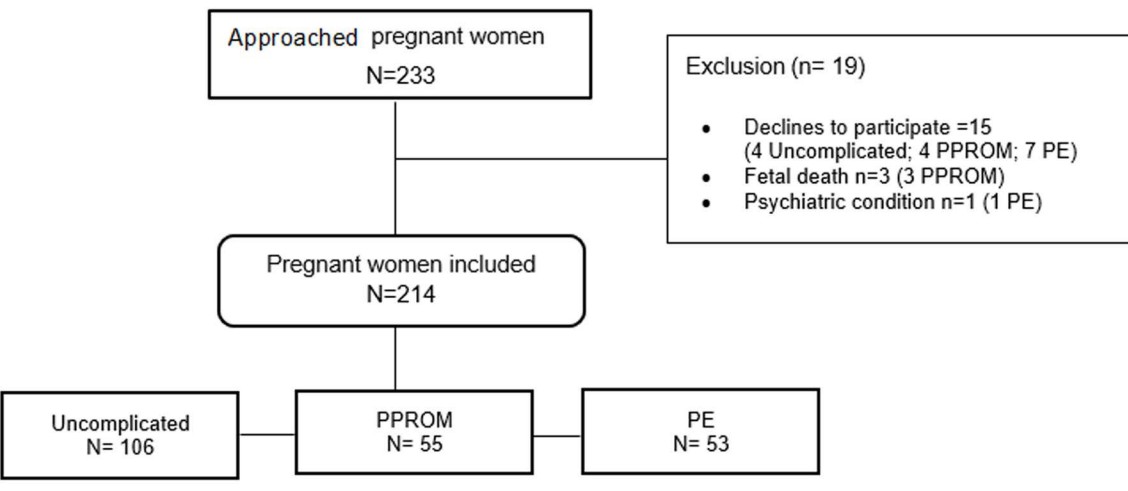

Reasons for exclusion and analyzed women by study group

**Fig 1. Sample recruitment flowchart.**

**Table 1. Baseline characteristics of the population.**

| | Total<br>n = 214 | Uncomplicated<br>n = 106 | PPROM<br>n = 55 | PE<br>n = 53 | P[1] |
|---|---|---|---|---|---|
| **Age (years)** | 34.9 (5.1) [19–46] | 35 (5.01) [23–45] | 35.4 (4) [27–43] | 34.1 (5.9) [19–46] | 0.391 |
| **Non-European origin** | 45 (21%) | 19 (17.9%) | 12 (21.8%) | 15 (28.3%) | 0.323 |
| **Low education** | 12 (5.8) | 5 (4.9) | 4 (7.4) | 3 (5.8) | 0.816 |
| **Unemployment** | 169 (79%) | 86 (81.1) | 45 (81.8) | 38 (74.7) | 0.324 |
| **Household** | | | | | |
| Coupled | 202 (94.4) | 102 (96.2) | 53 (96.4) | 47 (88.7) | 0.114 |
| Single | 12 (5.6) | 4 (3.8) | 2 (3.6) | 6 (11.3) | |
| **Primiparous** | 123 (57.5%) | 57 (53.8%) | 23 (41.8%) | 43 (81.1%) | <0.001 |
| **Previous miscarriage** | 53 (24.8) | 26 (24.5) | 16 (29.1) | 11 (20.8) | 0.603 |
| **Previous perinatal death** | 10 (4.7) | 1 (0.9) | 8 (14.5) | 1 (1.9) | <0.001 |
| **Smoker in pregnancy** | 16 (7.5) | 5 (4.7) | 5 (9.1) | 6 (11.3) | 0.286 |
| **Chronic hypertension** | 11 (5.1%) | 0 (0%) | 2 (3.6%) | 9 (17%) | <0.001 |
| **Dyslipidaemia** | 3 (1.4%) | 0 (0%) | 2 (3.6%) | 1 (1.9%) | 0.167 |
| **Cardiopathy** | 2 (0.9%) | 1 (0.9%) | 1 (1.8%) | 0 (0%) | 0.618 |
| **Diabetes** | | | | | |
| Pre-gestational | 8 (3.8) | 5 (4.7) | 0 | 3 (5.7) | 0.55 |
| Gestational | 26 (12.1) | 11 (10.4) | 9 (16.4) | 6 (11.3) | |
| **Gestational age at inclusion (days)** | 218 (18) [170–238] | 219 (18) [170–238] | 212 (20) [170–238] | 221 (16) [179–238] | 0.021 |
| **Gestational age at delivery (days)** | 252 (29) [164–293] | 277 (14) [164–293] | 226 (18) [174–281] | 230 (15) [191–257] | 0.001 |
| **Birthweight (gr)** | 2458 (910) [610–4640] | 3266 (418) [2140–4640] | 1754 (512) [720–3200] | 1622 (445) [610–4640] | 0.001 |
| **Fetal growth restriction** | 39 (18.2) | 4 (3.8) | 9 (16.4) | 26 (49.1) | <0.001 |
| **Caesarean delivery** | 95 (44.4%) | 22 (20.8%) | 29 (52.7%) | 44 (83%) | <0.001 |
| **Maternal hospitalisation (days)** | 4.2 (4.1) [0–27] | 2.6 (1.3) [0–12] | 7.7 (6.3) [2–27] | 3.5 (1.6) [0–10] | <0.001 |
| **Maternal hospitalisation in obstetric HDU** | 76 (35.5%) | 4 (3.8%) | 19 (34.5%) | 53 (100%) | <0.001 |
| **Maternal hospitalisation in obstetric HDU (days)** | 3 (5.3) [0–26] | 0.25 (0.9) [0–5] | 1.9 (3.1) [0–12] | 9.4 (6.5) [2–23] | <0.001 |

Mean (standard deviation) [min-max] or n (%), as appropriate

[1]Chi-squared or One-way ANOVA, as appropriate

PPROM: preterm premature rupture of membranes; PE: preeclampsia; HDU: High-Dependency Unit

groups, which remained significant after adjustment for gestational age at evaluation and STAI-T scores.

## Discussion

### Summary of key results

In this study, we investigated the impact of pregnancy complications during the third trimester on mental well-being, to test whether severe preeclampsia differentially adds risk. We identified a gradient of higher depression and anxiety across uncomplicated pregnancies, pregnancies at risk of prematurity due to preterm rupture of membranes and severe preeclampsia. It is noteworthy that women with severe preeclampsia exhibited the highest scores in anxiety, significantly different from those with preterm rupture of membranes.

Table 2. Anxiety and depression scores by study group.

| | Total n = 214 | Uncomplicated n = 106 | PPROM n = 55 | PE n = 53 | Trend tests | |
|---|---|---|---|---|---|---|
| | | | | | p[1] | p[2] |
| STAI-T | 14 (12) [1–43] | 12 (10) [2–36] | 16 (12) [5–43]* | 17 (17) [1–43] | 0.003 | – |
| STAI-S | 18 (14) [0–57] | 14 (9) [0–40] | 24 (16) [4–49] | 27 (17) [5–57]*,+ | <0.001 | <0.001 |
| EPDS | 7 (8) [0–23] | 4 (6) [0–23] | 8 (5) [0–18]* | 9 (5) [0–22]* | <0.001 | <0.001 |

Median (interquartile range) [min-max]

[1]Jonckheere–Terpstra trend test

[2]Adjusted by quantile regression by STAI-T and gestational age at evaluation (with polynomial linear contrast)

*Adjusted p value compared to the uncomplicated pregnancies < 0.05

+Adjusted p value compared to the PPROM pregnancies < 0.05

EPDS: Edinburgh Postnatal Depression Scale; STAI: State-Trait Anxiety Inventory; T: trait; S: state; PPROM: preterm premature rupture of membranes; PE: preeclampsia

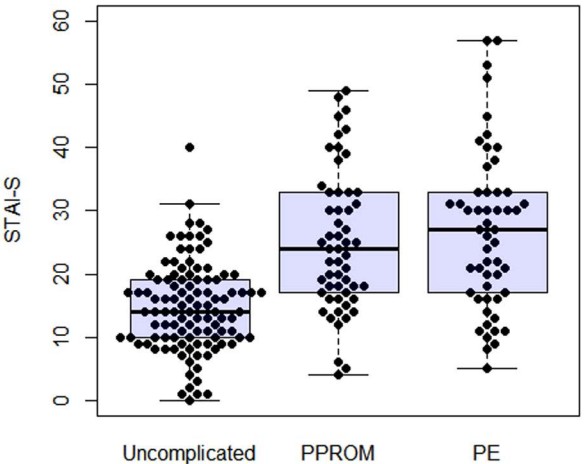

Scatterplot of observed scores with its boxplot:
median, 50th-75th percentile, and the limit for outliers (75th percentile plus 1.5 times the Interquartile range)

**Fig 2. Distribution of STAI-S scores by study group.**

## Comparison with previous literature

In our study, approximately 20% of women experienced anxiety, aligning with a recent meta-analysis that underscores a higher susceptibility to anxiety disorders during gestation compared to the postpartum period [3]. Overall, we found that 13.6% of women exhibited positive depression scores, surpassing the WHO's estimate of 10% for women with depression during pregnancy [31].

In the context of women with obstetric risk, our study finds support in a recent meta-analysis [14], that suggests that approximately one in three women hospitalised during pregnancy due to obstetric complications experiences clinical levels of depression or anxiety symptoms, which is double the prevalence in the general obstetric population.

Additionally, in line with our results, a cross-sectional study conducted by the Italian National Health Service [32], which included 2801 pregnant women, explored the risk of

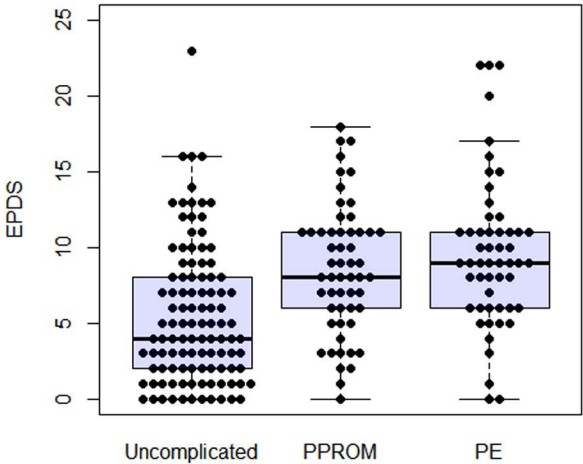

Scatterplot of observed scores with its boxplot:
median, 50th-75th percentile, and the limit for outliers (75th percentile plus 1.5 times the Interquartile range)

**Fig 3. Distribution of EPDS scores by study group.**

**Table 3. Abnormal Anxiety and depression scores by study group.**

| | Total n = 214 | Uncomplicated n = 106 | PPROM n = 55 | PE n = 53 | Trend tests | |
|---|---|---|---|---|---|---|
| | | | | | p[1] | p[2] |
| **STAI-T ≥30** | 20 (9.3) | 4 (3.8) | 5 (9.1) | 11 (20.8) | 0.001 | – |
| **STAI-S ≥30** | 45 (21) | 2 (1.9) | 19 (34.5)* | 24 (45.3)* | <0.001 | <0.001 |
| **EPDS ≥10** | 67 (31.3) | 20 (18.9) | 23 (41.8) | 24 (45.3)* | <0.001 | 0.001 |
| **EPDS ≥13** | 29 (13.6) | 10 (9.4) | 9 (16.4) | 10 (18.9) | 0.083 | 0.201 |

n (%)

[1]Linear-by-linear trend test

[2]Adjusted by logistic regression by gestational age at evaluation and STAI-T score, with polynomial linear contrast

*Adjusted p value compared to the uncomplicated pregnancies < 0.05

EPDS: Edinburgh Postnatal Depression Scale; STAI: State-Trait Anxiety Inventory; T: trait; S: state; PPROM: preterm premature rupture of membranes; PE: preeclampsia

anxiety and depression based on obstetric risk levels. It revealed that 28.9% scored ≥ 9 on the EPDS, and 17.1% scored ≥ 8 on the "Generalized Anxiety Disorder scale" (GAD-7). Obstetric risk groups showed an intermediate prevalence of anxiety and depression, with the high-risk group for fetal anomalies exhibiting the highest prevalence of anxiety (29.3%) and depression (49.1%).

Regarding the association between prematurity and maternal depression, there is conflicting evidence [17], probably reflecting that among women at risk of prematurity there is heterogeneity secondary to the specific complication that confers such risk. Our study adds to the limited evidence comparing the scores for depression and anxiety during pregnancy of women with severe preeclampsia to those complicated by PPROM. In line with our findings, Stramrood et al. [33] reported the prevalence of post-traumatic stress disorder to be higher in women with PE as compared to those with PPROM (21% vs. 14%).

The relationship between preeclampsia and postpartum depression has been more extensively explored in scientific literature [21,22]. Bergink et al. [23], in a cohort of 400,717 primiparous women reveal positive associations, particularly when preeclampsia is combined

with medical comorbidities, increasing the risk of psychiatric episodes in the first 3 months postpartum (TIR 4.81, 95% CI 2.72–8.50).

Caropreso et al., in a systematic review and meta-analysis (24), support this association by concluding that preeclampsia is linked to a 12% increase in the risk of postpartum depression. Our results extend this association to the antenatal period.

Possible explanations for this association, that are not mutually exclusive, include the impact of a common pathophysiological mechanism manifested as part of a systemic inflammatory response to the dysregulation of the immune system, which could lead to both preeclampsia and postpartum psychiatric episodes. Another plausible explanation suggests that the psychological impact resulting from a serious medical condition, either in the mother and/or the fetus, may increase the vulnerability of the woman to develop mental health disorders [20,34]. It could be hypothesised that the biological differential effect of preeclampsia on anxiety may have been attenuated in our cohort by the fact that all women were -per clinical protocol- under magnesium sulphate, with known sedative effects. Other factors that may influence our findings is the more frequent and longer stay of women with severe preeclampsia in the high-dependency unit, with more restricted autonomy than in a high-risk obstetric ward. Another directionally of our findings can be argued, since some evidence suggests that depression or anxiety may be risk factors for the onset of preeclampsia, mediated by endothelial and platelet dysfunction along with sympathetic hyperactivity, phenomena akin to cardiovascular disorders associated with depression [35–37].

## Research and clinical implications

Our study shows that women admitted for pregnancy complications at risk of preterm delivery experience higher levels of anxiety and depression. This emphasises the importance of early identification of anxiety and depression disorders, especially in women with obstetric risk, particularly those with severe preeclampsia. There is a need to optimise detection capabilities and to increase awareness and training among professionals regarding the importance of maternal mental health and its management during pregnancy. Health providers should provide a holistic approach to disease management and patient care, and nurses have a pivotal role in such an approach. Beyond treating physical symptoms, emotional, psychological, and social aspects of a patient's well-being should be addressed. Maternal lifestyle factors, including suboptimal nutrition and high levels of stress, may be associated with obstetric complications. This association is thought to potentially be mediated by effects on systemic and placental inflammation, oxidative stress, and cellular senescence, all involved in the pathophysiology of placental insufficiency [38]. By taking into account the whole person, nurses can better understand the root causes of illness and tailor care plans accordingly. The ideal method to evaluate the psychological aspects of the pregnant woman is through direct observation by health professionals. However, the use of screening instruments provides an efficient, objective, accessible, and easy-to-apply alternative in cases where clinical observation is not possible. An accurate assessment is key to guiding the implementation of appropriate therapeutic strategies. These strategies should be integrated into a multidisciplinary approach to ensure comprehensive care for mothers with obstetric complications. This has implications in staffing plans and resource allocation.

Early detection of perinatal depression and anxiety cannot be underscored, as it allows timely interdisciplinary approaches and interventions. Cognitive behavioural and interpersonal therapies are cost-effective during pregnancy. For major disorders, selective serotonin reuptake inhibitors (SSRIs) are first-line treatments for both depression and anxiety, meeting efficacy and safety standards during pregnancy and lactation. The use of SSRIs during

pregnancy has not been associated with an increased risk of preeclampsia or fetal growth restriction [39,40].

## Study limitations and strengths

The study has limitations, such as a sample recruited exclusively from a single centre, potentially impacting the generalizability of findings to other settings. Quantitively, our sample size did not allow further stratification of our population or further addressing confounding effects other than adjusting for gestational age at evaluation and STAI-Trait score. Qualitatively, we could have included a wider spectrum of pregnancy complications, although PE and PPROM account for the largest proportion of preterm maternal admissions to hospital. Moreover, our outcome measurements rely on self-reports, which may not entirely reflect a clinical diagnosis of depression or anxiety, as these scales are primarily used for screening, not comprehensive clinical diagnosis [41]. This may have resulted in biased interpretation of the results, including sampling and expectancy biases. Additionally, our study did not include a wider range of sociodemographic factors such as schooling and religious beliefs, as they can influence the psychological, cultural, and economic context of women with complicated pregnancies. We also grant some strengths to our study, as its prospective design make it more robust to biases. In addition, we included a control group of women with uncomplicated pregnancies and an intermediate group of women at risk of prematurity for reasons other than preeclampsia. Moreover, the measurements were performed within a short time interval within 72 hours of admission, enhancing the comparability between complicated pregnancies. Lastly, we adjusted for confounders, such as the baseline trait of anxiety and the gestational age at admission.

## Conclusion

Higher levels of anxiety and depression are present in women admitted to hospital for PPROM and severe PE. Severe PE differentially confers the highest risk. The importance of screening and offering specific interventions for patients with PE is highlighted.

## Author contributions

**Conceptualization:** Yolanda Giménez, Francesc Fatjó, Francesc Figueras, Angela Arranz.

**Data curation:** Yolanda Giménez, Francesc Fatjó.

**Formal analysis:** Francesc Figueras.

**Investigation:** Yolanda Giménez.

**Methodology:** Yolanda Giménez, Francesc Fatjó, Francesc Figueras.

**Project administration:** Yolanda Giménez.

**Resources:** Yolanda Giménez, Angela Arranz.

**Software:** Yolanda Giménez.

**Supervision:** Francesc Figueras, Angela Arranz.

**Validation:** Elena González, Francesc Fatjó, Aida Mallorquí, Sandra Hernández, Francesc Figueras, Angela Arranz.

**Visualization:** Yolanda Giménez.

**Writing – original draft:** Yolanda Giménez, Angela Arranz.

**Writing – review & editing:** Elena González, Francesc Fatjó, Aida Mallorquí, Sandra Hernández, Francesc Figueras, Angela Arranz.

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
