## [Decision Letter · Decision Letter 0]

15 Sep 2024

PONE-D-24-11220Anxiety and depression during pregnancy: differential impact in cases complicated by preeclampsia and preterm premature rupture of membranesPLOS ONE

Dear Dr. Figueras,

Thank you for submitting your manuscript to PLOS ONE. After careful consideration, we feel that it has merit but does not fully meet PLOS ONE’s publication criteria as it currently stands. Therefore, we invite you to submit a revised version of the manuscript that addresses the points raised during the review process.

We look forward to receiving your revised manuscript.

Kind regards,

Ochuwa Adiketu Babah, M.Sc.PH (Epidemiology), FWACS, FMCOG

Academic Editor

PLOS ONE

Journal Requirements:

2.  In the online submission form, you indicated that “The data underlying the results presented in the study are available from request to the corrresponding author”.

All PLOS journals now require all data underlying the findings described in their manuscript to be freely available to other researchers, either 1. In a public repository, 2. Within the manuscript itself, or 3. Uploaded as supplementary information. This policy applies to all data except where public deposition would breach compliance with the protocol approved by your research ethics board. If your data cannot be made publicly available for ethical or legal reasons (e.g., public availability would compromise patient privacy), please explain your reasons on resubmission and your exemption request will be escalated for approval.

Additional Editor Comments:

The authors should response to the comments of all the reviewers. They are advised to pay key attention to the comments of the Biostatistician. The revised manuscript shoudl be submitted with a point-by-point response to the reviewers' comments indicating clearly where changes were made.

Reviewers' comments:

Reviewer's Responses to Questions

**Comments to the Author**

1. Is the manuscript technically sound, and do the data support the conclusions?

Reviewer #1: Yes

Reviewer #2: Yes

Reviewer #3: Partly

2. Has the statistical analysis been performed appropriately and rigorously? 

Reviewer #1: Yes

Reviewer #2: Yes

Reviewer #3: No

3. Have the authors made all data underlying the findings in their manuscript fully available?

Reviewer #1: Yes

Reviewer #2: Yes

Reviewer #3: No

4. Is the manuscript presented in an intelligible fashion and written in standard English?

Reviewer #1: Yes

Reviewer #2: Yes

Reviewer #3: Yes

5. Review Comments to the Author

Reviewer #1: This is a sound, well-written and interesting manuscript on the impact of severe PE and PPROM on maternal anxiety and depression levels. ın my opinion the wellbeing of the mother depends on a multidisciplinary approach and supporting the maternal mental health is crucial. I have some minor recommendations for the authors:

1) What are the key strategies to support the mental health of high-risk pregnancies?

2) Could you please the safety and efficacy of medications like SSRIs during pregnancy and their impact on fetal growth.

3) The association between placenta related obstetric complications and maternal stress level should be discussed in more detail.

Reviewer #2: The topic is inexhaustible and little attended to in practice. The authors made a sound and straightforward approach, with inclusion and exclusion criteria that give strength to their research by purifying the sample to people without a previously recognized illness and without a history of violence: apparently they place the pathological event of anxiety and depression as a result of a complicated pregnancy as a trigger; however, I suggest that the authors include schooling and religious beliefs to strengthen the possible influence of these variables studied with the interpersonal, couple, cultural, gender, and economic context that must ultimately shake the identity of these women and that for clinicians are vital to understand these symptoms; the authors bias their study by ignoring other psychodynamics that must be addressed by the liaison psychiatry specialist, with psychotherapists and analysts and not only with nursing, which although it contributes, its psychosocial goodness requires this therapeutic team to alleviate the psychobiological aspects of depression and anxiety. Perhaps the authors could add this data to improve the sociodemographic aspects that could contribute to the anxious and depressive process that escapes their vision. This would make this article more complete and its recommendations more comprehensive.

Reviewer #3: Comments to the authors:

Title: Anxiety and depression during pregnancy: differential impact in cases complicated by preeclampsia and preterm premature rupture of membranes

The authors are trying to compare the three prospective cohort groups in terms of anxiety and depression during pregnancy. The objective of this study is to explore whether there is an increasing prevalence gradient of anxiety and depression among women with uncomplicated pregnancies, those admitted for PPROM, and those admitted for PE. The above objectives could be stated differently as it may be beneficial to compare the prevalence of the depression and anxiety among the three groups. Stated different, you could compare the prevalence of depression and anxiety between PPROM vs Control, PE vs Control, and perhaps PPROM vs PE.

Abstract:

The abstract is well written and there appears to be no significance difference between the groups: PPROM (27 vs. 24; p=0.049). Indeed, it is not clear if other covariate adjustments besides age have been done. It appears very little analysis result is presented in this section. Can you elaborate on the findings in the conclusion of the abstract?

Introduction

P4 l75-76: Rephrase to make this sound better. It is confusing in its current state.

P5l96-99: Break this to multiple sentences.

P5l100: remove “which”

Materials and design

Please include detail description of the different outcome variables.

L139: I believe the inclusion criteria had an upper band. Please describe this carefully. For example, women of childbearing ages 18yrs and older up to ….

Are the participants matched in terms of age at pregnancy or the number of prior births?

There may be several confounding variables that were overlooked.

The presentation of the study design is poor.

Study variables:

The two outcome variables: composite scores for the depression outcome and composite score for anxiety were reported. These were self-administered, meaning that there could be misinterpretation of the questions, social desirability bias, sampling bias.

It is also unclear how the if convenient samples were used.

It is unclear if any participant in study cohort A or B were allowed to choose to participate in the study or not. Elaborate on this.

Statistical analysis:

L192: “Qualitative variable” this should be replaced with categorical variable. It is impossible to characterized qualitative variable as frequency and percentages unless they are quantified.

The outcome variables are not normality distributed. It is unclear why the need to do the normality test using Shapiro Wilk’s test.

It is unclear how many times the measurement was taken for the pregnant women. Since this is a prospective cohort study, the assumption is that you follow the pregnant women until the event of interest occurs. Is there a baseline value? Were there no depression or anxiety at the onset? When did they develop depression and anxiety during the course of their pregnancy?

Please rewrite l195 -197. It is unclear if there is an inherent ordering of the group? If so, were you using a dichotomized outcome to do the trend test by Jonckheere–Terpstra?

Please clarify what dichotomization is been modelled and what model is been fitted.

It is unclear what the model/dichotomized outcome is been used to test the paired groups. Please clarify.

In the multivariate logistic model, again, the outcome variable is unclear. What is your dichotomization and why did you choose only gestational age and STAI-Trait score to be included in the model? Why were other variables collected not included in the model?

NOTE: The Chi-squared or One-way ANOVA methodology used to compare the groups does not appear to reveal what might be a critical point to consider. For example, a significant Anova results for the groups does not tell the reader which group differ. It only says that the at least one of the groups is different from the others.

Results:

L212-217 should be moved to study variable or material and design subsection

L217 should not have a period after PE as in “severe PE. Figure 1.”

When is baseline? What is the next follow-up time? If indeed there was a baseline, then it will be appropriate to model the change from baseline scores instead of what has been modeled.

What is the p-value p1 representing? Is it comparing the PPROM vs control or PPROM vs PE. It is not appropriate to do a three-way comparison and present p-values.

Suppose all p-values represents the comparison of PPROM vs Control group or PE vs Control group, then all significant p-values suggest you should adjust for those covariates in a multivariate model. This appears to have been omitted in the analysis.

The quantile regression results provide no explanation to what is going on. More covariates need to be explained in the model.

In fact, results are lacking and not sufficiently presented. Additionally, interpretation of the scanty tables presented are also lacking.

Summary and key results

It is unclear if the goal the authors set up is achieved in this instance. The summary of the key results was insufficient.

Again, the authors findings are unclear to say the least. If there are any findings, appropriate statistical methodology should be used to summarize the results. A discussion section should be used to explain what is going on. This is lacking in this study.

Research and clinical implications

The point is well noted, and your results and key points should be clearly pointed out in the result section and now supported in this section.

Study limitations and strengths

It does not appear you adjusted for confounders in the baseline covariates are state here:” Finally, we adjusted for confounders, 350 such as the baseline trait of anxiety and the gestational age at admission”.

Conclusion

Since the screening tools are not diagnostic tools, it is inappropriate to conclude using the results from this tool as a yard’s stick to classify prevalence of depression and anxiety among pregnant women ages 18 years and older.

Figure 1: This figure needs a title and a footnote description that makes it stand alone.

Figures 2 and 3: Needs a title and footnote description that makes it stand alone. The description should lead the reader to understand what they are looking at. For example, are the means/medians the same, smaller or larger? Is the variability similar or different?

6. PLOS authors have the option to publish the peer review history of their article (what does this mean? ). If published, this will include your full peer review and any attached files.

**Do you want your identity to be public for this peer review?** For information about this choice, including consent withdrawal, please see our Privacy Policy .

Reviewer #1: No

Reviewer #2: No

Reviewer #3: No

---

## [Author Response · Author response to Decision Letter 0]

19 Oct 2024

Journal Requirements:

1. Please ensure that your manuscript meets PLOS ONE's style requirements, including those for file naming

The citation of figures has been modified according to the requirements and the title of each figure has been added after the paragraph in which it is first cited, and its legend has been added.

2. In the online submission form, you indicated that “The data underlying the results presented in the study are available from request to the corrresponding author”

We have added the anonymized database as complementary material.

We have added the anonymized database as complementary material.

The ethical statement has been moved to the Methods section and the Ethical Considerations section has been removed.

Review Comments to the Author

Reviewer #1:

This is a sound, well-written and interesting manuscript on the impact of severe PE and PPROM on maternal anxiety and depression levels. In my opinion the wellbeing of the mother depends on a multidisciplinary approach and supporting the maternal mental health is crucial. I have some minor recommendations for the authors:

1. What are the key strategies to support the mental health of high-risk pregnancies?

We have added the following sentences in the Discussion section, in the section “Research and clinical implications”:

“This study emphasizes the importance of early identification of anxiety and depression disorders, especially in women with obstetric risk, particularly those with severe preeclampsia. There is a need to optimize detection capabilities and to increase awareness and training among professionals regarding the importance of maternal mental health and its management during pregnancy. Health providers should provide a holistic approach to disease management and patient care, and nurses have a pivotal role in such an approach. Beyond treating physical symptoms, emotional, psychological, and social aspects of a patient's well-being should be addressed. By taking into account the whole person, nurses can better understand the root causes of illness and tailor care plans accordingly. The ideal method to evaluate the psychological aspects of the pregnant woman is through direct observation by health professionals. However, the use of screening instruments provides an efficient, objective, accessible, and easy-to-apply alternative in cases where clinical observation is not possible. An accurate assessment is key to guiding the implementation of appropriate therapeutic strategies. These strategies should be integrated into a multidisciplinary approach to ensure comprehensive care for mothers with obstetric complications. This has implications in staffing plans and resource allocation.”

2. Could you please the safety and efficacy of medications like SSRIs during pregnancy and their impact on fetal growth.

This discussion extends beyond the framework of our study, but we have added a summarizing sentence in the Discussion section, in the section “Research and clinical implications”:

“Early detection of perinatal depression and anxiety cannot be underscored, as it allows timely interdisciplinary approaches and interventions. Cognitive behavioural and interpersonal therapies are cost-effective during pregnancy. For major disorders, selective serotonin reuptake inhibitors (SSRIs) are first-line treatments for both depression and anxiety, meeting efficacy and safety standards during pregnancy and lactation. The use of SSRIs during pregnancy has not been associated with an increased risk of preeclampsia or fetal growth restriction.”

Palmsten K, Huybrechts KF, Michels KB, Williams PL, Mogun H, Setoguchi S, et al. Antidepressant use and risk for preeclampsia. Epidemiology 2013; 24:682–91.

Malm H, Sourander A, Gissler M, Gyllenberg D, Hinkka-Yli-Salomäki S, McKeague IW, et al. Pregnancy complications following prenatal exposure to SSRIs or maternal psychiatric disorders: results from population-based national register data. Am J Psychiatry 2015; 172:1224–32.

3. The association between placenta related obstetric complications and maternal stress level should be discussed in more detail.

We have added a few sentences addressing this in the Discussion section, in the section “Research and clinical implications”:

“Maternal lifestyle factors, including suboptimal nutrition and high levels of stress, may be associated with obstetric complications. This association is thought to potentially be mediated by effects on systemic and placental inflammation, oxidative stress, and cellular senescence, all involved in the pathophysiology of placental insufficiency.”

Traylor CS, Johnson JD, Kimmel MC, Manuck TA. Effects of psychological stress on adverse pregnancy outcomes and nonpharmacologic approaches for reduction: an expert review. Am J Obstet Gynecol MFM. 2020;2(4):100229.

Reviewer #2:

The topic is inexhaustible and little attended to in practice. The authors made a sound and straightforward approach, with inclusion and exclusion criteria that give strength to their research by purifying the sample to people without a previously recognized illness and without a history of violence: apparently they place the pathological event of anxiety and depression as a result of a complicated pregnancy as a trigger; however, I suggest that the authors include schooling and religious beliefs to strengthen the possible influence of these variables studied with the interpersonal, couple, cultural, gender, and economic context that must ultimately shake the identity of these women and that for clinicians are vital to understand these symptoms; the authors bias their study by ignoring other psychodynamics that must be addressed by the liaison psychiatry specialist, with psychotherapists and analysts and not only with nursing, which although it contributes, its psychosocial goodness requires this therapeutic team to alleviate the psychobiological aspects of depression and anxiety. Perhaps the authors could add this data to improve the sociodemographic aspects that could contribute to the anxious and depressive process that escapes their vision. This would make this article more complete and its recommendations more comprehensive.

We agree on the importance of considering sociodemographic factors such as schooling and religious beliefs, as they can influence the psychological, cultural, and economic context of women with complicated pregnancies. In future research, it would be beneficial to include these variables to understand better the possible interactions between these factors and the onset of symptoms of anxiety and depression. However, in this study, we decided to focus on obstetric and mental health factors directly associated with complicated pregnancy, and therefore we have not exhaustively covered other psychosocial aspects that are undoubtedly relevant. The study’s ethical approval would not cover further data collection without a previous major amendment. We have added a sentence on the study limitations acknowledging this.

“… as these scales are primarily used for screening, not comprehensive clinical diagnosis [38]. Additionally, our study did not include a wider range of sociodemographic factors such as schooling and religious beliefs, as they can influence the psychological, cultural, and economic context of women with complicated pregnancies.”

Regarding the therapeutic approach, we agree that an interdisciplinary approach, including psychiatrists, psychotherapists, and other specialists, is essential to treat maternal anxiety and depression. We have added a sentence highlighting this:

“Early detection of perinatal depression and anxiety cannot be underscored, as it allows timely interdisciplinary approaches and interventions. Cognitive behavioral and interpersonal therapies are cost-effective during pregnancy.”

Reviewer #3:

Comments to the authors:

Title: Anxiety and depression during pregnancy: differential impact in cases complicated by preeclampsia and preterm premature rupture of membranes

1. The authors are trying to compare the three prospective cohort groups in terms of anxiety and depression during pregnancy. The objective of this study is to explore whether there is an increasing prevalence gradient of anxiety and depression among women with uncomplicated pregnancies, those admitted for PPROM, and those admitted for PE. The above objectives could be stated differently as it may be beneficial to compare the prevalence of the depression and anxiety among the three groups. Stated different, you could compare the prevalence of depression and anxiety between PPROM vs Control, PE vs Control, and perhaps PPROM vs PE

In addition to the trend analysis (that was the primary analysis), paired comparisons were also performed, as stated in the Methods section. The results of these analyses correspond to the symbols * (PPROM vs. uncomplicated and PE vs. uncomplicated) and + (PE vs PROM) in Tables 2 and 3.

2. Abstract:

The abstract is well written and there appears to be no significance difference between the groups: PPROM (27 vs. 24; p=0.049). Indeed, it is not clear if other covariate adjustments besides age have been done. It appears very little analysis result is presented in this section. Can you elaborate on the findings in the conclusion of the abstract?

This difference corresponds to the adjusted quantile regression of medians comparing PE and PPROM, adjusted for gestational age at evaluation and STAI-Trait, as detailed in “Statistical analysis” of the Methods section.

The abstract conclusion has been modified to address the reviewer's point:

“Higher levels of anxiety and depression are present in women admitted to the hospital for PPROM and severe PE. Compared to PPROM, severe PE is associated with higher scores of anxiety. The importance of screening ...”

3. Introduction

3.1. P4 l75-76: Rephrase to make this sound better. It is confusing in its current state.

The sentence has been reworded:

“Depression and anxiety during pregnancy, with a prevalence between 10% and 20%, have a major impact on maternal and perinatal morbidity and mortality”

has been replaced by:

“Depression and anxiety during pregnancy, with a prevalence of 10% to 20%, significantly contribute to maternal and perinatal morbidity and mortality”.

3.2. P5l96-99: Break this to multiple sentences.

We have replaced:

“Among the reasons for hospitalization, preeclampsia (PE) and preterm premature rupture of membranes (PPROM) stand out as the most frequent, with a significant impact on maternal and neonatal morbidity and mortality, accounting for 15% and 30% of cases of prematurity, respectively”

by

“Among the reasons for hospitalization, preeclampsia (PE) and preterm premature rupture of membranes (PPROM) stand out as the most frequent. They carry a significant impact on maternal and neonatal morbidity and mortality, accounting for 15% and 30% of the preterm deliveries, respectively.”

3.3. P5l100: remove “which

Done.

4. Materials and design

4.1. Please include detail description of the different outcome variables.

The description of the main outcome variables already extends to 30 lines of the manuscript (lines 146-176).

Regarding the control variables, we have added more detailed definitions.

…, primiparous (no previous delivery beyond 20 weeks); previous miscarriage (pregnancy loss before 20 weeks); previous perinatal death (beyond 20 weeks); smoker in pregnancy (1 or more cigarettes/day at booking); chronic hypertension (blood pressure >140/90 measured on two occasions more than 12h apart before 20 weeks of pregnancy); dyslipidaemia (pre-pregnancy cholesterol > 200 mg/dL); maternal heart disease (congenital structural anomalies, acquired ischemic, valvular or arrhythmic conditions); …; fetal growth restriction (an estimated fetal weight below the 10th percentile according local standards); ...

4.2 L139: I believe the inclusion criteria had an upper band. Please describe this carefully. For example, women of childbearing ages 18yrs and older up to ….

We checked the inclusion criteria, an only gestational age at recruitment has an upper limit (from 26+0 to 33+6 weeks). Maternal age had no upper limit.

4.3. Are the participants matched in terms of age at pregnancy or the number of prior births?

The study groups were not matched. See our answer to the reviewer’s points 4.5.

4.4. There may be several confounding variables that were overlooked.

We admit this limitation. Our sample size did not allow us to adjust for more than 2 potential confounders and only gestational age at evaluation and STAI-Trait score were considered. See a more technical address in response to query 6.7. We added a sentence in the section on the study limitations acknowledging this point:

“Quantitively, our sample size did not allow further stratification of our population or further addressing confounding effects other than adjusting for gestational age at evaluation and STAI-Trait score”.

4.5. The presentation of the study design is poor.

We thank the reviewer for this pertinent comment. Upon consultation with our statistician, we have amended taxonomically the description of the study design in both the Abstract and Methods sections:

“A cross-sectional t study was conducted involving three groups of pregnant women consecutively attended.”

5. Study variables:

5.1. The two outcome variables: composite scores for the depression outcome and composite score for anxiety were reported. These were self-administered, meaning that there could be misinterpretation of the questions, social desirability bias, sampling bias.

This limitation is already acknowledged in the paragraph on the study limitations. We have extended the discussion according to the reviewer's points.

“Moreover, our outcome measurements rely on self-reports, which may not entirely reflect a clinical diagnosis of depression or anxiety, as these scales are primarily used for screening, not comprehensive clinical diagnosis [38]. This may have resulted in biased interpretation of the results, including sampling and expectancy biases.”

5.2. It is also unclear how the if convenient samples were used. It is unclear if any participant in study cohort A or B were allowed to choose to participate in the study or not. Elaborate on this.

All women gave informed consent to participate. Figure 1 details that 4 uncomplicated women, 4 with PPROM, and 7 with PE declined to participate. To further clarify it we have modified figure 1 and replaced “Selected pregnant women” by “Approached pregnant women”.

6. Statistical analysis:

6.1. L192: “Qualitative variable” this should be replaced with categorical variable. It is impossible to characterized qualitative variable as frequency and percentages unless they are quantified.

Done.

6.2. The outcome variables are not normality distributed. It is unclear why the need to do the normality test using Shapiro Wilk’s test.

Shapiro-Wilk tests were done to for

---

## [Decision Letter · Decision Letter 1]

23 Dec 2024

PONE-D-24-11220R1Anxiety and depression during pregnancy: differential impact in cases complicated by preeclampsia and preterm premature rupture of membranesPLOS ONE

Dear Dr. Figueras,

Thank you for submitting your manuscript to PLOS ONE. After careful consideration, we feel that it has merit but does not fully meet PLOS ONE’s publication criteria as it currently stands. Therefore, we invite you to submit a revised version of the manuscript that addresses the points raised during the review process.

We look forward to receiving your revised manuscript.

Kind regards,

Ochuwa Adiketu Babah

Academic Editor

PLOS ONE

Journal Requirements:

Additional Editor Comments :

Dear Author,

I adjudge your responses to the reviewers’ comments generally satisfactory. I will appreciate further revision before final decision is made on the manuscript.

I could find the following comments which you mentioned you have included in the manuscript with the corresponding reference for the first comment. As a guide, these were basically responses to editorial and reviewer 2 comments. Please ensure this is done and respond point-by-point indicating the line numbers where the additions are made.

1. Early detection of perinatal depression and anxiety cannot be underscored, as it allows timely interdisciplinary approaches and interventions. Cognitive behavioural and interpersonal therapies are cost-effective during pregnancy. For major disorders, selective serotonin reuptake inhibitors (SSRIs) are first-line treatments for both depression and anxiety, meeting efficacy and safety standards during pregnancy and lactation. The use of SSRIs during pregnancy has not been associated with an increased risk of preeclampsia or fetal growth restriction.”

2. … as these scales are primarily used for screening, not comprehensive clinical diagnosis [38]. Additionally, our study did not include a wider range of sociodemographic factors such as schooling and religious beliefs, as they can influence the psychological, cultural, and economic context of women with complicated pregnancies.

3. Early detection of perinatal depression and anxiety cannot be underscored, as it allows timely interdisciplinary approaches and interventions. Cognitive behavioral and interpersonal therapies are cost-effective during pregnancy.

Please include these comments in the manuscript and re-submit for review and final decision.

Thank you.

Best regards.

D

Reviewers' comments:

Reviewer's Responses to Questions

**Comments to the Author**

1. If the authors have adequately addressed your comments raised in a previous round of review and you feel that this manuscript is now acceptable for publication, you may indicate that here to bypass the “Comments to the Author” section, enter your conflict of interest statement in the “Confidential to Editor” section, and submit your "Accept" recommendation.

Reviewer #1: All comments have been addressed

Reviewer #3: All comments have been addressed

2. Is the manuscript technically sound, and do the data support the conclusions?

Reviewer #1: Yes

Reviewer #3: Yes

3. Has the statistical analysis been performed appropriately and rigorously? 

Reviewer #1: Yes

Reviewer #3: Yes

4. Have the authors made all data underlying the findings in their manuscript fully available?

Reviewer #1: Yes

Reviewer #3: Yes

5. Is the manuscript presented in an intelligible fashion and written in standard English?

Reviewer #1: Yes

Reviewer #3: Yes

6. Review Comments to the Author

Reviewer #1: (No Response)

Reviewer #3: I have reviewed the resubmitted revisions, and the authors have made a commendable effort in addressing my comments. Overall, the revisions adequately respond to the feedback provided and the manuscript reads significantly better. I have no further comments at this time.

7. PLOS authors have the option to publish the peer review history of their article (what does this mean? ). If published, this will include your full peer review and any attached files.

**Do you want your identity to be public for this peer review?** For information about this choice, including consent withdrawal, please see our Privacy Policy .

Reviewer #1: No

Reviewer #3: No

---

## [Author Response · Author response to Decision Letter 1]

27 Dec 2024

Find itemized answers to each of the Editor’s queries. There are no further comments from the reviewers to be addressed.

Journal Requirements:

The reference list is complete. The only reference that has been removed was former number 39, that corresponded to the Helsinki Declaration. As proposed by the Editorial office we moved all the content that was in a section headed “Ethical considerations” to the Methods section. Because the Ethical Approval code is now provided, we felt it was not necessary to keep this reference.

In addition, after the first revision in response to query#2 of reviewer #1 we included 3 new references (lines 504-515 of the clean copy of the manuscript and lines 470, 473 and 475 of the changes-annotated pdf).

Additional Editor Comments:

Dear Author,

I adjudge your responses to the reviewers’ comments generally satisfactory. I will appreciate further revision before final decision is made on the manuscript. I could find the following comments which you mentioned you have included in the manuscript with the corresponding reference for the first comment. As a guide, these were basically responses to editorial and reviewer 2 comments. Please ensure this is done and respond point-by-point indicating the line numbers where the additions are made.

1. Early detection of perinatal depression and anxiety cannot be underscored, as it allows timely interdisciplinary approaches and interventions. Cognitive behavioural and interpersonal therapies are cost-effective during pregnancy. For major disorders, selective serotonin reuptake inhibitors (SSRIs) are first-line treatments for both depression and anxiety, meeting efficacy and safety standards during pregnancy and lactation. The use of SSRIs during pregnancy has not been associated with an increased risk of preeclampsia or fetal growth restriction.”

The whole paragraph is already included (lines 359-365 of the clean manuscript and line 335 in the changes-annotated pdf). Supporting these sentences there are 2 new references (#39 and #40).

Now it reads:

“Early detection of perinatal depression and anxiety cannot be underscored, as it allows timely interdisciplinary approaches and interventions. Cognitive behavioural and interpersonal therapies are cost-effective during pregnancy. For major disorders, selective serotonin reuptake inhibitors (SSRIs) are first-line treatments for both depression and anxiety, meeting efficacy and safety standards during pregnancy and lactation. The use of SSRIs during pregnancy has not been associated with an increased risk of preeclampsia or fetal growth restriction [39-40].”

2. … as these scales are primarily used for screening, not comprehensive clinical diagnosis [38]. Additionally, our study did not include a wider range of sociodemographic factors such as schooling and religious beliefs, as they can influence the psychological, cultural, and economic context of women with complicated pregnancies.

The whole paragraph is already included (lines 376-381 of the clean manuscript and line 344 in the changes-annotated pdf).

Now it reads:

“…as these scales are primarily used for screening, not comprehensive clinical diagnosis [41]. This may have resulted in biased interpretation of the results, including sampling and expectancy biases. Additionally, our study did not include a wider range of sociodemographic factors such as schooling and religious beliefs, as they can influence the psychological, cultural, and economic context of women with complicated pregnancies.”

3. Early detection of perinatal depression and anxiety cannot be underscored, as it allows timely interdisciplinary approaches and interventions. Cognitive behavioral and interpersonal therapies are cost-effective during pregnancy.

This has been addressed above in point #1.

Now it reads:

“Early detection of perinatal depression and anxiety cannot be underscored, as it allows timely interdisciplinary approaches and interventions. Cognitive behavioural and interpersonal therapies are cost-effective during pregnancy. For major disorders, selective serotonin reuptake inhibitors (SSRIs) are first-line treatments for both depression and anxiety, meeting efficacy and safety standards during pregnancy and lactation. The use of SSRIs during pregnancy has not been associated with an increased risk of preeclampsia or fetal growth restriction [39-40].”

---

## [Decision Letter · Decision Letter 2]

20 Feb 2025

PONE-D-24-11220R2Anxiety and depression during pregnancy: differential impact in cases complicated by preeclampsia and preterm premature rupture of membranesPLOS ONE

Dear Dr. Figueras,

Thank you for submitting your manuscript to PLOS ONE. After careful consideration, we feel that it has merit but does not fully meet PLOS ONE’s publication criteria as it currently stands. Therefore, we invite you to submit a revised version of the manuscript that addresses the points raised during the review process.

 Thank you for your patience whilst we assessed your revised manuscript. To improve the reporting of the manuscript we have some minor revision requests and questions to help clarify the context of the study. In the methods section please could you include the following information:

Details of the sample size calculation / power calculation used.Additional detail on the recruitment methods that were used.Could you please clarify whether this is a primary study or whether this is a re-analysis of data collected as part of a broader study / a secondary or sub-study? If this is the case please include these details in the manuscript text.

We look forward to receiving your revised manuscript.

Kind regards,

Emma Campbell, Ph.D

Staff Editor

PLOS ONE

Reviewers' comments:

Reviewer's Responses to Questions

**Comments to the Author**

1. If the authors have adequately addressed your comments raised in a previous round of review and you feel that this manuscript is now acceptable for publication, you may indicate that here to bypass the “Comments to the Author” section, enter your conflict of interest statement in the “Confidential to Editor” section, and submit your "Accept" recommendation.

Reviewer #1: All comments have been addressed

2. Is the manuscript technically sound, and do the data support the conclusions?

Reviewer #1: Yes

3. Has the statistical analysis been performed appropriately and rigorously? 

Reviewer #1: Yes

4. Have the authors made all data underlying the findings in their manuscript fully available?

Reviewer #1: Yes

5. Is the manuscript presented in an intelligible fashion and written in standard English?

Reviewer #1: Yes

6. Review Comments to the Author

Reviewer #1: The authors have conducted the recommended revisions, the manuscrpt may be accepted in its present form.

7. PLOS authors have the option to publish the peer review history of their article (what does this mean? ). If published, this will include your full peer review and any attached files.

**Do you want your identity to be public for this peer review?** For information about this choice, including consent withdrawal, please see our Privacy Policy .

Reviewer #1: No

---

## [Author Response · Author response to Decision Letter 2]

28 Feb 2025

Journal Requirements:

In the methods section please could you include the following information:

1. Details of the Sample Size Calculation / Power Calculation Used

We expected to recruit 50 women with preterm premature rupture of membranes (PPROM) and 50 with preeclampsia (PE) over the study period. An additional 100 uncomplicated pregnancies were intended to be included as a control group, resulting in a total planned sample size of 200 participants. To determine the adequacy of our sample size for detecting a trend in proportions across the study groups, we performed a power analysis using the linear-by-linear trend test. We assumed a significance level of 0.05 and three ordered exposure groups with expected proportions of 5%, 20%, and 30%. Using a trend test framework with equally spaced exposure levels, the estimated power for detecting a significant linear trend was 85.8%.

We added this paragraph to the Methods section in a revised version.

2. Additional Details on the Recruitment Methods Used

Women with complications requiring admission to the high-dependency unit of a referral hospital were recruited consecutively during the study period (December 2016–December 2018). Pregnant women without medical or obstetric complications, who attended routine antenatal care at the same institution during the study period, were recruited simultaneously alongside patients with complications.

We have accordingly reworded the paragraph in the Methods section where the recruitment is detailed.

3. Clarification on the Nature of the Study

Our study was a primary study. However, after the first submission of our manuscript, a study analyzing a smaller subgroup of patients (n=168) who had postpartum follow-up was conducted and published in the Spanish journal Atención Primaria (“Aten Primaria. 2024 Sep 25;57(3):103085. doi: 10.1016/j.aprim.2024.103085”). The data and analyses in the current manuscript were not included in that secondary study.

---

## [Editor Report · Decision Letter 3]

10 Mar 2025

Anxiety and depression during pregnancy: differential impact in cases complicated by preeclampsia and preterm premature rupture of membranes

PONE-D-24-11220R3

Dear Dr. Figueras,

We’re pleased to inform you that your manuscript has been judged scientifically suitable for publication and will be formally accepted for publication once it meets all outstanding technical requirements.

Kind regards,

Marianne Clemence

Staff Editor

PLOS ONE
---

## [Editor Report · Acceptance letter]

PONE-D-24-11220R3

PLOS ONE

Dear Dr. Figueras,

I'm pleased to inform you that your manuscript has been deemed suitable for publication in PLOS ONE. Congratulations! Your manuscript is now being handed over to our production team.

Kind regards,

on behalf of

Dr Marianne Clemence

Staff Editor

PLOS ONE